# Development and Application of an FSI Model for Floating VAWT by Coupling CFD and FEA

Wenping Luo [1,2], Weiqin Liu [1,2,*], Shuo Chen [2], Qilu Zou [2] and Xuemin Song [1,2]

[1] Key Laboratory of High-Performance Ship Technology, Wuhan University of Technology, Ministry of Education, Wuhan 430063, China; luowenping@whut.edu.cn (W.L.); sxmxs@whut.edu.cn (X.S.)

[2] Department of Naval Architecture and Ocean Engineering, School of Naval Architecture, Ocean and Energy Power Engineering, Wuhan University of Technology, Wuhan 430063, China; c19563784@163.com (S.C.); zouqiluzql@163.com (Q.Z.)

[*] Correspondence: liuweiqin_123@sina.com

**Abstract:** The emerging floating vertical axis wind turbines (VAWTs) are regarded as a preferred solution to overcome the challenges faced by the traditional horizontal type in open-sea environments. Numerous numerical models have been advanced for assessing this novel object. However, current fully coupled models predominantly rely on simplified theories, assuming a linear fluid load and a one-dimensional slender beam structure. Despite computational fluid dynamic and finite element (CFD-FEA) coupling being qualified for high precision, this technology remains limited to the fixed VAWT field. To predict the load and structural response accurately and comprehensively, this study aims to extend CFD-FEA technology to floating VAWTs. First, an aero-hydro-moor-elastic fully coupled model is developed, and this model is validated by comparing it with several model experiments. Subsequently, a full-scale floating straight-bladed VAWT is simulated with the geometry and numerical models introduced. Furthermore, load and structural responses in a typical case are analyzed in both time and frequency domains. Finally, the sensitivity analysis of each structure part in floating VAWTs to environmental parameters is conducted and discussed. The discovery highlights the intricate nature of tower structural response, which incorporates 2-node, 3-node, wind frequency, and wave frequency components. Distinct from blades or floating foundations, which are primarily influenced by a single environmental parameter, the tower response is significantly amplified by the combined effects of wind and waves.

**Keywords:** floating vertical axis wind turbine; structural response; computational fluid dynamic; finite element; fluid-structure interaction

## 1. Introduction

Wind energy, characterized by its abundant availability and ease of development, has emerged as the fastest-growing and most successfully commercialized renewable energy source, with a cumulative installed capacity of 743 GW by 2020 [1]. Nonetheless, the progression towards large-scale and deep-sea deployment of floating horizontal axis wind turbines (HAWTs), which is aimed at reducing unit costs and increasing total capacity, has led researchers to increasingly confront some serious challenges [2].

Vertical axis wind turbines (VAWTs) are deemed an optimal solution to address these concerns, capitalizing on the numerous benefits summarized in the literature [3]. Additionally, Tzanakis [4] and SeaTwirl [5] have also indicated that floating VAWTs exhibit superior performance in dense wind farms compared to conventional HAWTs. Consequently, numerous renowned concepts have been developed, including DeepWind [6], NOVA, VertAx [7], VertiWind [8], and INFLOW [9], as illustrated in Figure 1. However, it should be highlighted that VAWTs possesses a unique structural configuration compared to HAWTs (i.e., two blade-inflow interactions and dynamic attack angle), resulting in significant changes in their load and structural characteristics. When integrated with a floating

platform, floating VAWTs would encounter a more intricate mechanism in the coupled wind-wave environment, which remains unexplored and holds significant interest.

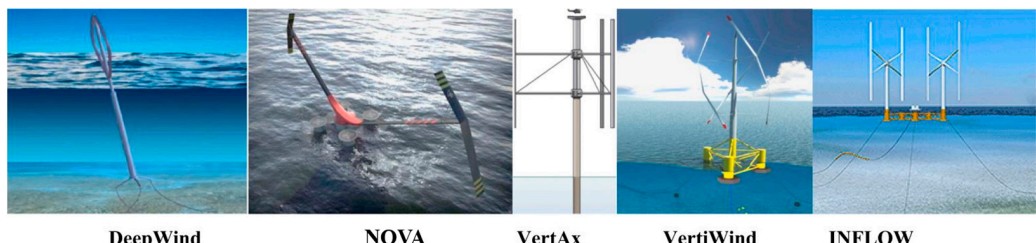

**Figure 1.** Current famous designs and concepts for offshore floating VAWT.

Fluid-structure interaction (FSI) analysis is indispensable for the study of floating VAWTs, given the intricate load and structural response scenarios. It is a well-established fact that floating wind turbines constitute a unique rigid-flexible multi-structure, characterized by the slenderness and flexibility of both the blade and the tower, in conjunction with a more rigid floating foundation. Under the complex wind-wave coupling loadings, these floating wind turbines demonstrate significant FSI characteristics. Consequently, it is crucial to address load and structural response in an integrated manner for accuracy. Currently, there are two primary FSI methods in offshore wind technology: the simplified fully coupled model and the direct computational fluid dynamics-finite element analysis (CFD-FEA) coupling approach.

The first is the fully coupled aero-hydro-moor-elastic model, where finite element analysis or multi-body dynamics is integrated into the existing simplified load coupled model. In 2007, Larsen [10] at the Technical University of Denmark initially presented a comprehensive HAWC2 model for floating VAWTs. This model utilized the actuator cylinder (AC) theory, Morrison theory, and multi-body dynamics theory for its aerodynamic, hydrodynamic, and structural solution, respectively. Subsequently, Wang [11] from the Norwegian University of Science and Technology advanced a novel coupling model for floating VAWTs, referred to as SIMO-RIFLEX-DMS. The key contribution of this model resides in its improvement in precision for each module through the integration of double-multiple streamtube (DMB) model, potential flow theory, and finite element beam model. Furthermore, Cheng [12] replaced the aerodynamic module with the AC model, resulting in the formation of SIMO-RIFLEX-AC. In summary, although these coupling models demonstrate enhanced computational efficiency, their application is primarily confined to the initial design phase for offshore VAWTs, given that each module is constructed from simplified representations.

The second approach involves the utilization of the direct CFD-FEA coupling method, wherein all fluid loads are resolved by solving the N-S equations and transferred to the structural model in real time, facilitating the computation of the structural response at each tine step. Zhang et al. [13] employed the CFD-FEA coupling method to conduct numerical simulations of a three-blade VAWT, aiming to optimize its structure. Based on this method, Liu and Xiao [14] conducted an investigation into the effects of blade flexibility on the performance and structural safety of VAWTs with straight blades, concluding that soft blades (low stiffness) are not beneficial for aerodynamic performance and safety. Similarly, Borouji [15] conducted a fatigue analysis for NREL-5MW fixed wind turbines. Achieving CFD-FEA coupling within the ANSYS platform, Marzec [16] conducted an FSI analysis of the Savonius VAWT and determined that minor blade deformations could enhance power output by approximately 5%, attributing a considerable impact to centrifugal force on structural deformation. A similar study was also conducted by Hoerner [17]. It is evident that CFD-FEA coupling has emerged as a novel research approach in the field of wind turbines given that CFD can effectively account for the effects of viscosity and nonlinearity on fluid loads, while FEA excels in analyzing the structural response of the entire system. However, current research remains confined to fixed VAWTs.

As previously mentioned, although simplified fully coupled models are capable of conducting integrated FSI analysis for floating VAWTs, they are only suitable for initial design due to their neglect of nonlinear aerodynamic loads and the response of actual three-dimensional structures. The direct CFD-FEA coupling method demonstrates remarkable precision; however, it remains confined to the domain of fixed VAWTs. To comprehensively and accurately predict the performance of floating VAWTs, this study aims to extend the CFD-FEA coupling method to the field of floating VAWTs. This paper is organized as follows. The numerical modelling process and theory of CFD-FEA are first introduced in Section 2, forming an aero-hydro-moor-elastic model. Subsequently, in Section 3, each module of the developed model is individually validated by comparing its results with available experimental data. Section 4 then describes the geometry and numerical model of a floating straight-bladed VAWT, and mesh-dependent modal analyses and simulated cases are also included. After that, the acquired external load and structural responses in a typical case are analyzed in both time and frequency domains in Section 5. Further, Section 6 discusses the effect of wind speed, wave height, and wind-wave misaligned angles on a floating VAWT system. Finally, several key findings are presented in Section 7. This study presents a precise FSI numerical method for evaluating floating VAWTs, particularly benefiting the detailed design stage.

## 2. Numerical Modelling

The authors first attempt to couple CFD and FEA to solve hydroelasto-plastic problems and set a simple ship as a case study [18]. Then, this method is tried for extension in the floating VAWT area. Due to the difficulty in coupling technology for complex multiple-freedom systems, the load and performance of floating VAWTs were first investigated aero-hydro-mooring load coupling was achieved [19], but the structural dynamic was not included. Hence, the present work attempts to extend CFD-FEA into floating VAWT application to form a fully coupled model considering structural dynamics. Some key and fundamental theories are described here.

### 2.1. Fully Coupled Model

The fully coupled model, as illustrated in Figure 2, primarily comprises an aerodynamic module, a hydrodynamic module, a mooring module, a multi-body motion module, and a finite element module. In the aerodynamic module, the size of the rotation domain (where the grids have the same rotation as the rotor) and background domain (where the grids are stationary) is initially determined based on the imported rotor model. Subsequently, the mesh is divided using the finite volume method, and the rotating motion of the mesh is implemented through the sliding mesh technique. Finally, the unsteady Reynold average Navier-Stokes (RANS) equation is solved in conjunction with the wind field to obtain the aerodynamic behavior of the rotor. A similar process is followed in the hydrodynamic module, with the free movement of the floating foundation being achieved through overset technology (where the whole grids in motion domain have the same movement as the floating VAWT, and the data transfer between the background and the overset is achieved by overset boundary interpolation).

The presence of a gas-liquid two-phase flow throughout the flow field and a free surface necessitates the application of the Euler multiphase flow and VOF methods to generate two-phase flows and capture their interfaces (coupling aero- and hydro-dynamics). In the mooring module, the load at the anchor point is determined by solving the quasi-static catenary equation within the mooring module. Subsequently, the aerodynamic load, hydrodynamic load, and mooring load are combined into the dynamic response module, and the dynamic fluid body interaction (DFBI) equation is solved by considering the structure's inherent parameters (weight, center of gravity position, moment of inertia, etc.). The six-degree-of-freedom (6-DOF) motion response of the entire floating system is derived, and the superimposed 1-DOF rotational motion is further resolved on this foundation.

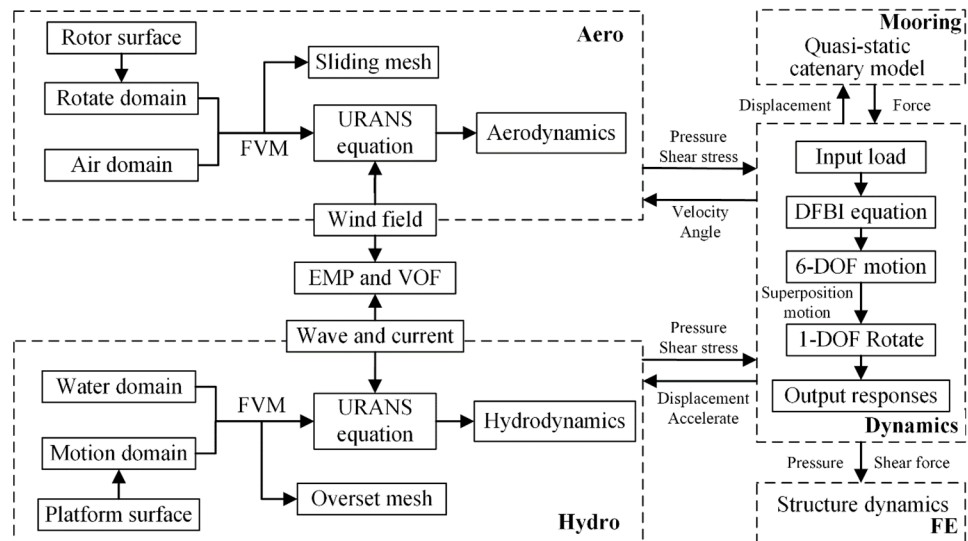

**Figure 2.** The developed fully coupled FSI model for offshore floating VAWTs.

Concurrently, after each calculation step, the rotor velocity and angle provided by the dynamic response module are returned to the aerodynamic module, the platform displacement and acceleration are relayed to the hydrodynamic module, and the mooring displacement is fed back to the mooring module to update the position and facilitate the subsequent calculation step. During each calculation step, all loads are mapped to the structural model via the fluid-structure coupling interpolation function, and the structural dynamic equation is solved utilizing the dynamic implicit algorithm. Ultimately, the structural response at each step, encompassing stress and displacement, is obtained. This fully coupled model is developed by integrating STAR-CCM+17.06.008-R8 and ABAQUS 2023. Details about the CFD-FEA settings and data transition can be found in the author's previous study [18]. This paper aims to extend the coupling scheme into a floating VAWT area.

*2.2. Aero-Hydrodynamics*

Fluid load is obtained for both aerodynamic and hydrodynamic modules by solving the Navier-Stokes (N-S) equations. CFD has the advantage of directly integrating the governing equations by considering both aerodynamic and hydrodynamic forces. The N-S equation, being a transient equation at every moment, poses challenges for computer applications. To address this, the URANS method assumes that the variables in the equation consist of an average quantity and a perturbation quantity. Consequently, the momentum equations based on the average concept are presented as follows:

$$\rho\left[\frac{\partial \overline{u_i}}{\partial t} + \frac{\partial \overline{(u_i u_j)}}{\partial x_j}\right] = -\frac{\partial \overline{p}}{\partial x_i} + \mu \nabla^2 u_i + \frac{\partial}{\partial x_j}(-\rho \overline{u_i' u_j'}) \tag{1}$$

where $u$ is velocity, $\rho$ is density of fluid, $p$ is press, $\mu$ is molecular viscosity, $\overline{u}$ is average speed of time, $\overline{p}$ is average pressure of time, and $(-\rho \overline{u_i' u_j'})$ is an item of Reynolds stress. As new unknowns come in, a turbulence model should be used to close the equations. Particularly, the approach combining the RANS equation and the k-ω turbulence model has been validated in calculating aerodynamics and hydrodynamics [20,21]. Transport equations of turbulent kinetic energy and dissipation rate for the SST k-ω model are described as follows:

$$\frac{\partial}{\partial t}(\rho k) + \frac{\partial}{\partial x_i}(\rho k u_i) = \frac{\partial}{\partial x_j}\left(\left(\mu + \frac{\mu_t}{\sigma_k}\right)\frac{\partial k}{\partial x_j}\right) + \widetilde{G_k} - Y_k + S_k \tag{2}$$

$$\frac{\partial}{\partial t}(\rho\omega) + \frac{\partial}{\partial x_i}(\rho\omega u_i) = \frac{\partial}{\partial x_j}\left(\left(\mu + \frac{\mu_t}{\sigma_\omega}\right)\frac{\partial\omega}{\partial x_j}\right) + G_\omega - Y_\omega + D_\omega + S_\omega \tag{3}$$

where $\widetilde{G_k}$ is a term of turbulent kinetic energy generation for $k$, $G_\omega$ is a term of dissipation rate generation, $D_\omega$ is a term of cross diffusion for $\omega$, $Y_k$ is a term of turbulent kinetic energy dissipation for $k$, $Y_\omega$ is a term of turbulent kinetic energy dissipation for $\omega$, and $S_k$ and $S_\omega$ are user-defined terms. Then, the basic solution of the flow field can be obtained by combining Equations (1)–(3). To capture the gas-liquid two-phase interface precisely, volume of fluid (VOF) is adopted, in which a phase volume fraction is used to ascertain the phase distribution and the location of the interface, which is defined as follows:

$$\delta_i = \frac{V_i}{V} \tag{4}$$

where $V_i$ is volume of phase $i$ in the mesh element, and $V$ is the volume of the mesh element. The sum of the volume fractions of all phases in an element must be one. According to the value of the volume fraction, the existence of different phases or fluids in grid cells can be distinguished. $\delta_i = 0$ means that the corresponding mesh element has no phase at all, $\delta_i = 1$ means that the corresponding mesh element is completely phase filled, and $0 < \delta_i < 1$ means that there are interfaces of multiple phases.

### 2.3. Structural Dynamics

The basic equation of structural dynamics is shown below, and since damping exists in the actual case, it is usually added in the form of Rayleigh damping:

$$[M]\{\ddot{u}\} + [C]\{\dot{u}\} + [K]\{u\} = \{F(t)\} \tag{5}$$

$$[C] = \alpha[M] + \beta[K] \tag{6}$$

where the first item to the left is inertia force, the second and third items are both internal forces, and the right item is external loads. $M$, $C$, and $K$ are the mass, damping, and stiffness matrixes, respectively; $u$ is the node displacement matrix; and $\alpha$ and $\beta$ are the damping coefficients. Due to the long calculation time of the whole fluid-structure coupling and the input of transient loads in each time step, the implicit dynamic algorithm is used to solve the equation. The implicit dynamic algorithm adopts the Newmark method for iterative calculation, and the calculation process is shown in [18].

### 2.4. Mooring Model

A floating vertical shaft fan system requires a mooring system to limit the drift motion, and the catenary is suspended between two points (anchor point and guide point) under its own gravity. In order to consider the influence of the mooring system on the hydrodynamic force and motion, the solution is based on the quasi-static catenary model. Figure 3 shows the forces and curves in this model. Its shape is defined as follows:

$$x = \frac{c}{\lambda_0 g}u + \frac{c^2}{DL_{eq}\lambda_0 g}\sinh(u) + \psi \tag{7}$$

$$y = \frac{c}{\lambda_0 g}\cosh(u) + \frac{c^2}{2DL_{eq}\lambda_0 g}\sinh^2(u) + \zeta \tag{8}$$

$$c = \frac{\lambda_0 L_{eq}g}{\sinh(u_2) - \sinh(u_1)} \tag{9}$$

where $g$ is the gravity acceleration, and $\lambda_0$ and $L_{eq}$ are the mass per unit length and relaxation length of the catenary, respectively. $D$ is the stiffness of the catenary; $\psi$ and $\zeta$

are integration constants depending on the position of the two end points and the total catenary mass, respectively; and $u$ is related to inclination angle $\phi$ as follows:

$$\tan \phi = \sinh(u) \tag{10}$$

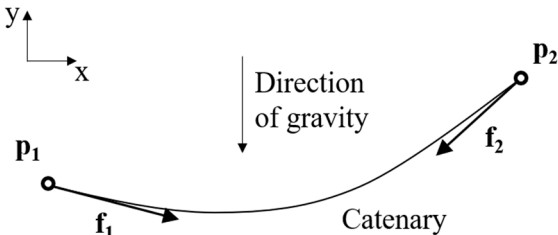

**Figure 3.** Illustration of load and position in a catenary model.

The parameter values $u_1$ and $u_2$ represent the positions of the catenary end points $p_1$ and $p_2$, respectively, in parameter space. Furthermore, the force and acting on the two end points can be expressed by $u_1$ and $u_2$, respectively.

$$f_{1,x} = c, \quad f_{1,y} = c\sinh(u_1) \tag{11}$$

$$f_{2,x} = -c, \quad f_{1,y} = -c\sinh(u_2) \tag{12}$$

## 3. Validation

To validate the efficacy of this model in assessing the load and structural response of a floating vertical axis wind turbine, it is crucial to adopt the concept of decoupled validation for each module. Consequently, the validity of the aerodynamic module, hydrodynamic module, and the fluid-structure coupling of the CFD and FEA modules will be examined in the subsequent sections of this chapter. It should be noted that the validation of the CFD-FEA coupling scheme for offshore structures has been conducted in previous studies [18]. To ensure the feasibility of this method in predicting floating VAWTs, only aerodynamic and hydrodynamics are demonstrated here.

### 3.1. Aerodynamic Validation

In 2015, Maeda [22] conducted load tests on VAWTs within the wind tunnel of Mie University and examined the impact of blade number on its aerodynamic performance. During this experiment, the wind speed remained constant at 8 m/s. The tested model was a small, straight-bladed vertical axis impeller with a radius of 1 m, a vertical blade height of 1.2 m, an NACA0021 airfoil type, a chord length of 0.265 m, and an included blade number of 2, 3, 4, and 5. As the primary objective of this study is to investigate a three-blade VAWT, the three-blade model was selected for comparison. It is worth noting that numerous researchers have also conducted numerical simulations of this standardized test, such as LEI [23] using the IDDES model. Consequently, this section compares the simulation outcomes with the experimental and IDDES results.

Figure 4 illustrates the variations in the average power coefficient of the VAWT due to different tip speed ratios (TSRs). It can be discerned that the power coefficient of this VAWT type initially increases and then decreases with the TSR. The wind energy utilization rate attains its maximum when the TSR equals 1.78. Furthermore, the values and altering trends of the three data exhibit consistency. Additionally, the power coefficient of a single blade is shown under the working condition of TSR = 1.78. It can be observed that the peak value of aerodynamic load occurs when the blade is situated in the upwind region, while the amplitude and fluctuation are minimal in the downwind region. Consequently, a single peak value characteristic is manifested in a rotation period. Overall, the present

aerodynamic load aligns closely with the results of experimental studies and pertinent literature, thereby corroborating the effectiveness of the aerodynamic module.

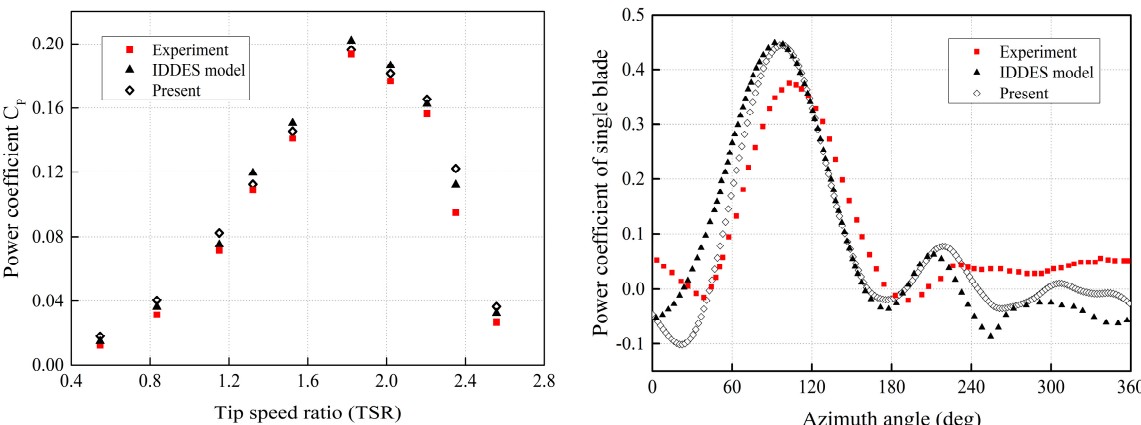

**Figure 4.** Aerodynamic comparison of the experiment, available numerical data, and present model.

Compared to the conventional simplified model, this approach benefits from the consideration of nonlinear effects. Furthermore, it elucidates the blade vortex field and tail vortex during impeller rotation and provides a brief analysis of their characteristics. As illustrated in Figure 5, the eddy field of the blade varies at different azimuths. It can be discerned that two vortices are generated at the trailing edge of the blade at 0°, while almost no vortices are formed near the blade's 90° position. A significant increase in vortex generation is observed as the blade enters the downwind area. At 180°, one vortex is formed at the middle and another at the trailing edge of the blade, respectively, while at 270°, multiple vortices are generated from the blade's head to tail. As vorticity signifies the boundary layer separation and dynamic stall, it also elucidates the reason for the calculated aerodynamic performance being optimal at approximately 90°. Observing the VAWT vorticity in the wake region, large eddy shedding is presented at the upper, top edges and the trailing edge, and then flows backward. Traversing the downwind region, the pre-existing eddies are once again perturbed, giving rise to a multitude of chaotic eddies in the wake.

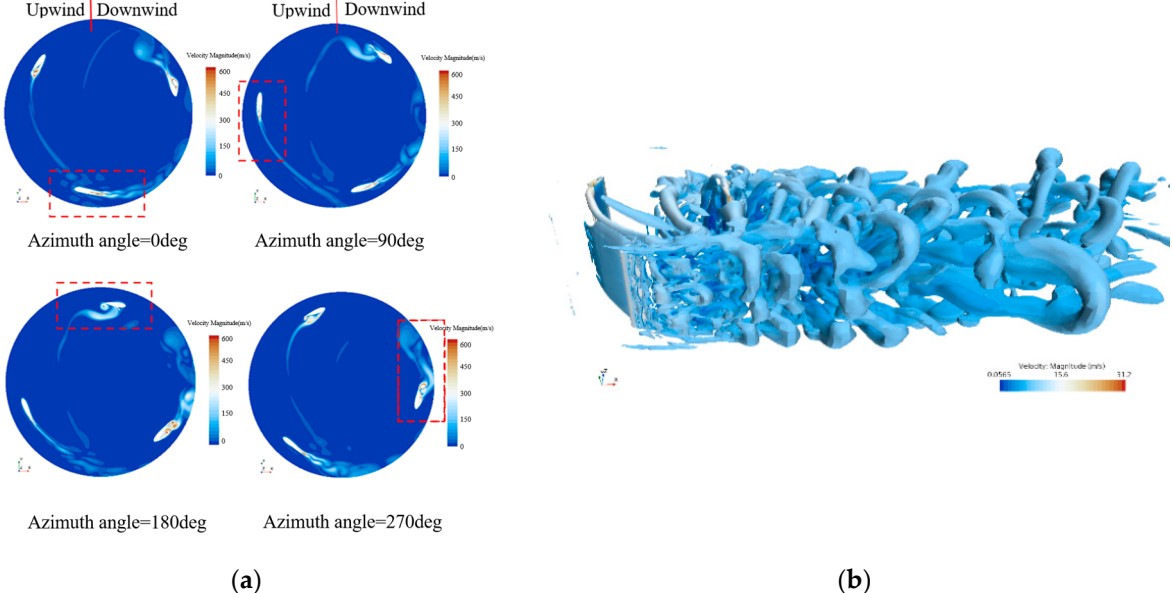

**Figure 5.** The included aerodynamic nonlinearities in this model: (**a**) blade eddy where the blade in red box is the target; (**b**) wake vortex.

*3.2. Hydrodynamic Validation*

In recent years, the National Renewable Energy Laboratory (NREL) has advanced a series of international collaborative projects concerning floating wind turbines. These projects include the execution of pool tests for Spar-type floating foundations [24] and semi-submersible floating platforms [25]. Notably, the DeepCWind semi-submersible platform within the OC4 project is currently acknowledged as an outstanding platform. Consequently, numerous scholars have conducted numerical analyses on this platform. Thus, this paper presents hydrodynamic verification for free decay and regular wave tests.

The simulation of the free decay motion for the three primary degrees of freedom of the platform is conducted, yielding the periods of pitch, roll, and heave. The obtained results are compared with experimental outcomes and other numerical studies, as illustrated in Table 1. The analysis reveals that the discrepancy between the motion period and test data in this study is less than 5%, thereby meeting the specified accuracy requirements. The hydrodynamic performance under regular wave conditions is subsequently analyzed, and the results are compared with Peng [26], who conducted the same simulation using WAMIT. Figure 6 displays the time-history curves of pitch and heave. The results of the pitch and heave motion under the action of regular waves exhibit good agreement.

Overall, the motion response obtained under the consideration of mooring influence aligns well with the experimental results in both free attenuation and regular wave conditions, thereby corroborating the effectiveness of the hydrodynamic module in the model.

**Table 1.** Period comparison of free decaying motion.

| DOF. | Exp [27] | Simo/Riflex [28] | Fast V8.1 [29] | Unsteady CFD [30] | Naoe-FOAM-SJTU [31] | Present |
|------|----------|------------------|----------------|-------------------|---------------------|---------|
| Pitch | 26.8 | 25.8 | 25.0 | 25.2 | 25.8 | 25.6 |
| Roll | 26.9 | 26.0 | 25.0 | 25.3 | – | 25.7 |
| Heave | 17.5 | 17.1 | 17.7 | 17.8 | 17.58 | 17.7 |

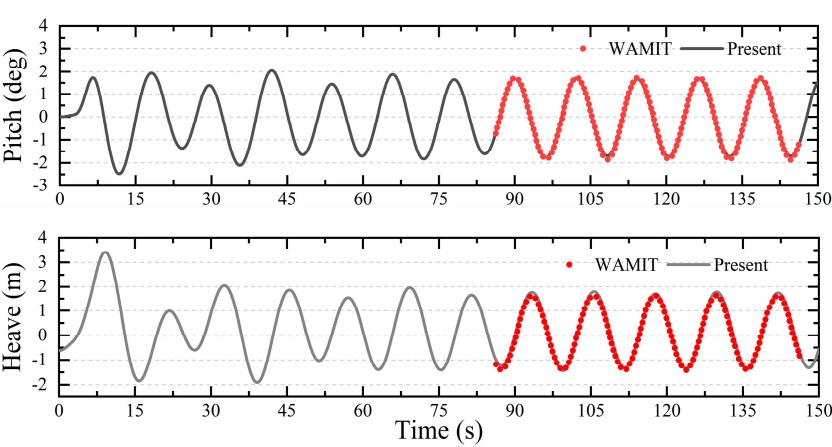

**Figure 6.** Comparison of hydrodynamic performance predicted by WAMIT and the present model.

## 4. Model Description

### 4.1. OF-VAWT System

The floating VAWT remains in the early stages of development, with no successful commercial prototype currently in operation. Consequently, a mature floating VAWT design, which was previously published by the author, is utilized as the analysis object in this study. As depicted in Figure 7, the VAWT is composed of three straight blades, the floating foundation is constructed from three buoy platforms, and the wind turbine is stabilized by three mooring cables. Table 2 illustrates the structural properties of the whole system. The detailed properties of rotor, platform, and mooring are presented in [19].

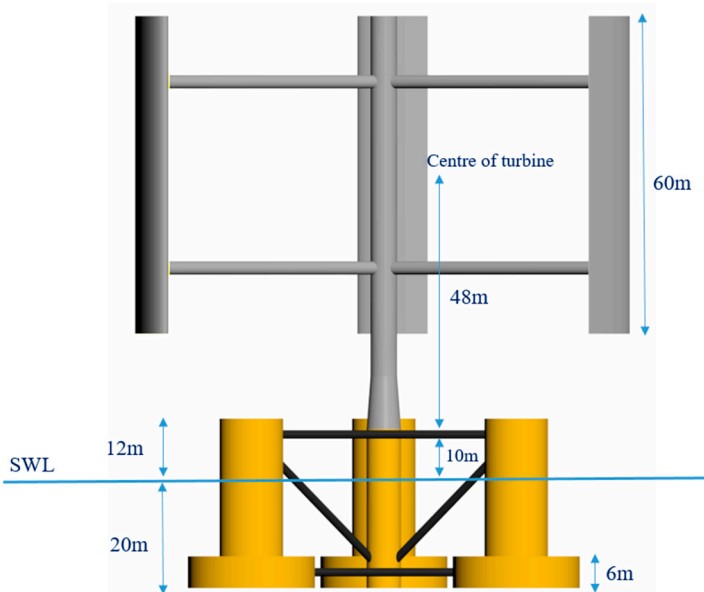

**Figure 7.** Illustration of the floating straight-bladed VAWT.

**Table 2.** Properties of this system.

| Property | Parameter |
|---|---|
| Mass | $1.37 \times 10^7$ kg |
| CM location below SWL | 12.53 m |
| Total structure roll inertia bout CM | $8.6 \times 10^9$ kg·m² |
| Total structure pitch inertia bout CM | $8.6 \times 10^9$ kg·m² |
| Total structure yaw inertia bout CM | $1.226 \times 10^{10}$ kg·m² |

### 4.2. CFD Model

The complex multiple degree-of-freedom (DOF) motion of the floating VAWT system, including its overall 6-DOF free motion and the 1-DOF rotating motion of the rotor, requires the establishment of a rotate domain that can accommodate the entire rotor, followed by the creation of an overset domain that can encompass the entire floating system. As illustrated in Figure 8, the detailed boundary conditions of the entire computational domain are configured as follows: The inlet serves as the velocity inlet, the outlet and top function as the pressure outlet, and the sides as well as the bottom are designated as symmetry. To facilitate the grid simulation of intricate motion, two sets of overset grids are employed, resulting in the external surface of the rotation domain and the overset domain being established as overset. In order to replicate the influence of a catenary on the motion of the floating VAWT, parameters such as the position of the fairlead and anchored points, as well as the stiffness of the mooring line, are incorporated. Subsequently, the mooring load at different time intervals is calculated based on the quasi-static catenary model, thereby restricting the drift motion.

The mesh generation strategy employed in this study is visualized in Figure 9 as a longitudinal section. To accurately capture the waveshape during wave propagation, volume refinement technology is employed to refine the X and Z directions of the free liquid surface region. Additionally, taking into account the effects of the wake disturbed by the rotor, appropriate volume refinement of the wake region is necessitated. Notably, a consistent mesh size near the interface of the two regions should be maintained to avoid errors associated with data interpolation.

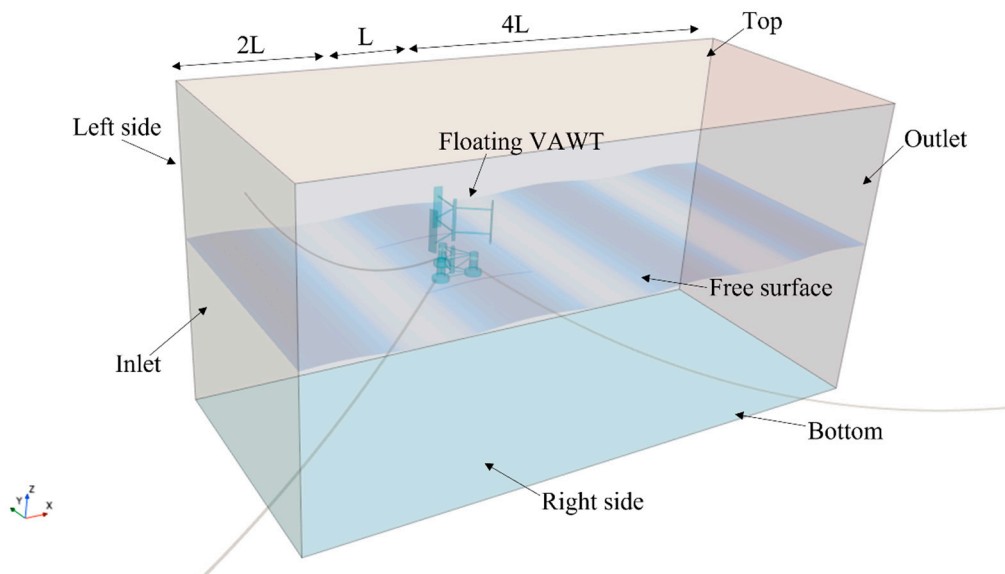

**Figure 8.** The established computational domain for this floating VAWT.

After establishing the computational domain and mesh strategy, the mesh size is determined through grid independence analysis. The floating VAWT is subjected to the combined effects of uniform wind and regular waves, resulting in complex gas-liquid two-phase flow and energy dissipation during wind-wave propagation. Consequently, the waveshape in the propagation can be utilized as the object for grid independence analysis. Based on the variation in grid basic size, the meshes can be categorized into Mesh1, Mesh2, and Mesh3, with corresponding total mesh numbers of 520 w, 902 w, and 1415 w, respectively. Subsequently, the waveshape obtained for each mesh type are compared with the theoretical values. Table 3 compares the discrepancies between the numerical results and the theoretical values under various mesh divisions. It is observed that Mesh2 and Mesh3 exhibit higher accuracy and meet the required standards. Finally, Mesh2 is selected as the grid scheme in this study, taking into account both computational efficiency and accuracy. The total grid number stands at 902 w.

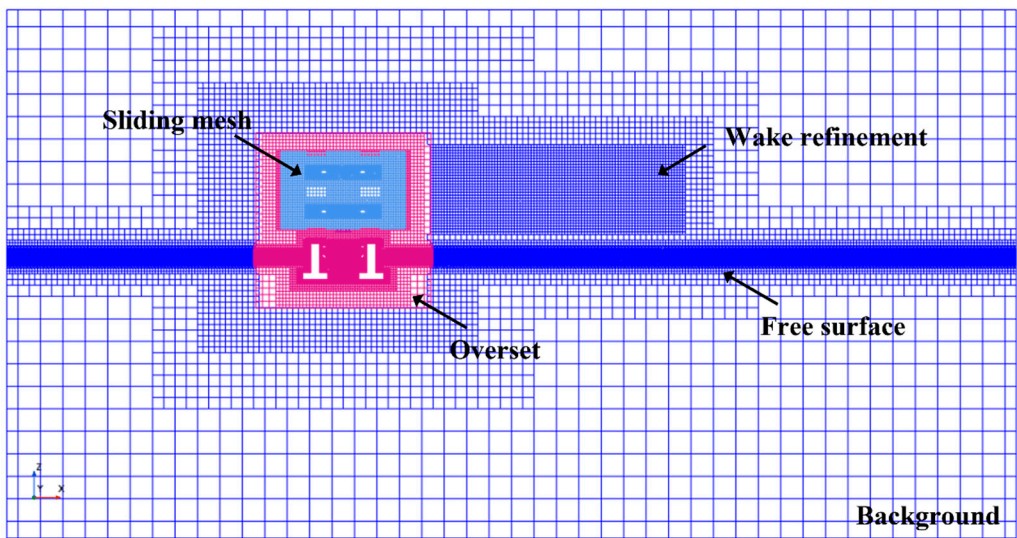

**Figure 9.** Illustration of the mesh dividing strategy in a longitudinal section.

**Table 3.** Comparison of wave results under different mesh types.

| Type | Grid Number | Numerical Value | Theoretical Value | Error |
|---|---|---|---|---|
| Mesh1 | 520 w | 2.67 m | 3 m | 11% |
| Mesh2 | 902 w | 2.88 m | 3 m | 4% |
| Mesh3 | 1415 w | 2.92 m | 3 m | 2.67% |

### 4.3. FE Model

Figure 10 illustrates the established finite element structural model, primarily composed of quadrilateral shell elements (S4R), with a limited application of triangular shell elements (S3) employed to facilitate a smooth transition at sharp corners. The blade structure, fabricated from glass fiber composite material, exhibits a density of 1900 kg/m$^3$, an elastic modulus of 20 GPa, and a Poisson ratio of 0.13. The tower structure, constructed from No. 35 steel, possesses a density of 7800 kg/m$^3$, an elastic modulus of 200 GPa, and a Poisson ratio of 0.26. The struts of the rotor are crafted from No. 45 steel, characterized by a density of 7850 kg/m$^3$, an elastic modulus of 205 GPa, and a Poisson ratio of 0.29. Lastly, the semi-submersible platforms are fabricated entirely from Q235 steel, with a density of 7850 kg/m$^3$, an elastic modulus of 210 GPa, and a Poisson's ratio of 0.3. The basic mesh size is established at 0.3 m, followed by the refinement of the blade area with significant curvature. Ultimately, the total mesh count amounts to 60 w.

The motion of the floating structure is continuously resolved in this numerical approach via the CFD module, with the sole requirement being the determination of the structure deformation under wind and wave load in the structure solver. Consequently, displacement constraints must be incorporated to restrict the rigid body displacement. Referring to the DNV specification [32], boundary conditions are established at the junctions of the three mooring cables with the platform, as illustrated in Figure 10. The following settings are employed: X = Y = Z = 0 at the point BC1 nearest to the wave's direction, Y = Z = 0 at BC2, and Z = 0 at BC3.

To facilitate the subsequent analysis of the structural response, a modal analysis of the entire structure under dry mode conditions is conducted, resulting in the extraction of the first four modes and their corresponding frequencies, as detailed in Table 4. The frequencies of the first modes of the structure are predominantly within the lower range, suggesting that the overall stiffness of the structure is relatively flexible, especially for the tower.

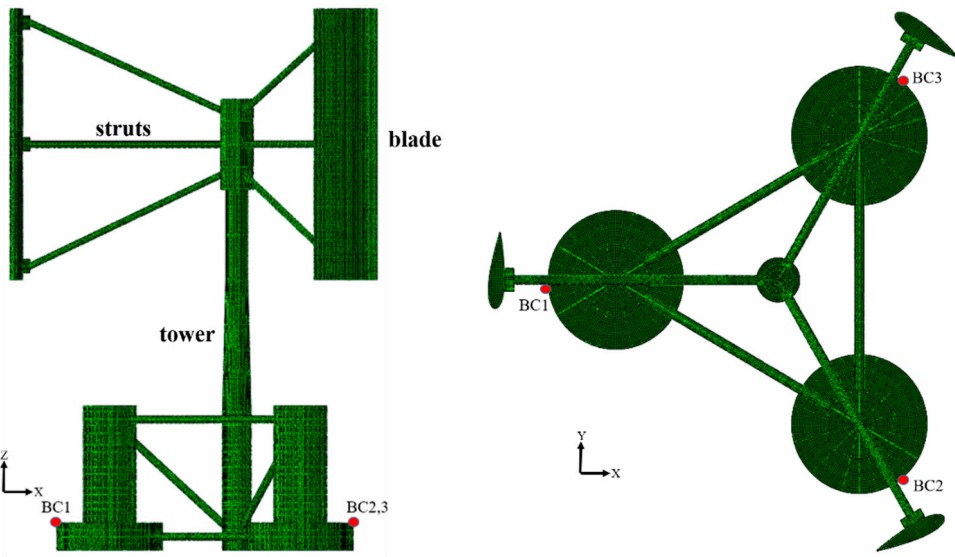

**Figure 10.** The established FE model and boundary settings.

**Table 4.** Modal results of the floating VAWT.

| Modal | Frequency (Hz) | Behavior |
|---|---|---|
| First order | 0.40435 | Tower 2-node |
| Second order | 0.46483 | Blade translation |
| Third order | 1.5446 | Tower 3-node |
| Fourth order | 1.9768 | Struts 2-node |

*4.4. Environmental Cases*

The primary focus of this study is on the effects of wind and waves, with the flow load being omitted due to its predominant influence on drag. The floating VAWT system designed in this chapter is intended for deployment in the South China Sea. Consequently, the wind speed and wave height are determined based on the environmental statistics. A series of loading cases are developed according to the principles of variable wind speed, variable wave height, and variable wind-wave misaligned angle, as detailed in Table 5. LC1, LC2, LC3, and LC4 represent the variable wind speed group; LC2, LC5, LC6, and LC7 define the variable wave height group; and LC2, LC8, LC9, and LC10 define the variable wind-wave misaligned angle group. Notably, LC2 serves as a public loading case and is thereby considered a typical case. It should be noted that the control system maintains the rotor in an optimal working state by adjusting its speed. Consequently, the rotor is given a fixed rotate speed in this study to keep within the optimal TSR of 1.75 under any wind speed. (Refer to [19] for studies determining the optimal tip ratio.)

**Table 5.** Loading cases of the floating VAWT under the wind-wave condition.

| Case | WS (m/s) | RV (rpm) | WH (m) | WF (Hz) | MA (°) |
|---|---|---|---|---|---|
| LC.1 | 10 | 3.342 | 3 | 0.133 | 0 |
| LC.2 (typical) | 15 | 5.013 | 3 | 0.133 | 0 |
| LC1.3 | 20 | 6.685 | 3 | 0.133 | 0 |
| LC.4 | 25 | 8.356 | 3 | 0.133 | 0 |
| LC.5 | 15 | 5.013 | 1 | 0.133 | 0 |
| LC.6 | 15 | 5.013 | 6 | 0.133 | 0 |
| LC.7 | 15 | 5.013 | 9 | 0.133 | 0 |
| LC.8 | 15 | 5.013 | 3 | 0.133 | 30 |
| LC.9 | 15 | 5.013 | 3 | 0.133 | 60 |
| LC.10 | 15 | 5.013 | 3 | 0.133 | 90 |

LC: loading case; WS: wind speed; RV: rotate velocity; WH: wave height; WF: wave frequency: MA: misaligned angle.

## 5. Results

To reveal the load and structural response characteristics of a floating VAWT in wind and wave environments, time-domain and frequency-domain analyses are performed on the acquired data. Given the constraints of space, the results from LC2, a typical case, are presented in this section.

Figure 11 depicts the operational status of the floating system within the wind-wave environment. It is evident that the entire machine experiences a slight pitch, with the wave run-up along the fore-side column. Additionally, the rotating VAWT generates partial eddies at the upper and bottom edges of the blades. Furthermore, as illustrated by the wind speed cloud image along the longitudinal section, an evident low-speed wake zone is formed. The wake length has a direct impact on the wind farm layout.

Given that time-domain load and motion responses have been presented and analyzed in previous work [19], the frequency-domain analysis is extended and addressed in this section. Figure 12 displays the frequency-domain curves of the torque and lateral force generated by the rotor, alongside the frequency-domain curves of the pitch and heave motion generated by the entire floating system. In this typical case, the wave frequency is 0.133 Hz, and the rotor rotation frequency is 0.083 Hz. The maximum value of the

aerodynamic load is observed to occur near 0.252 Hz, which is approximately three times (3P) the rotation frequency. This is attributed to the alternating appearance of the three equivalent blades, resulting in three peak characteristics per cycle. The primary source of energy for the aerodynamic load is derived from the wind load. Moreover, a small peak value near the wave frequency of 0.133 Hz is noticed, suggesting that the aerodynamic load may also be influenced by waves.

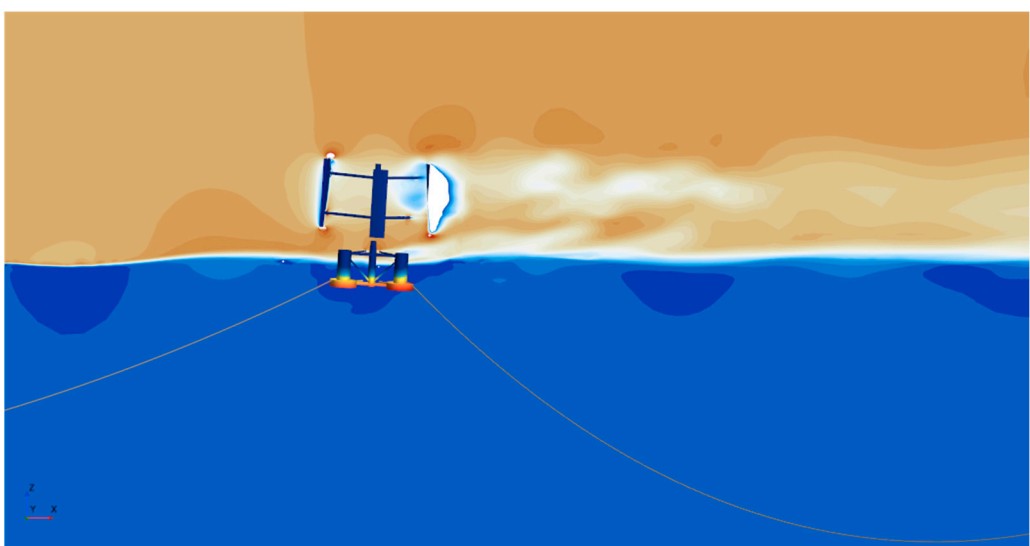

**Figure 11.** Operating status of a floating vertical axis fan in wind and waves.

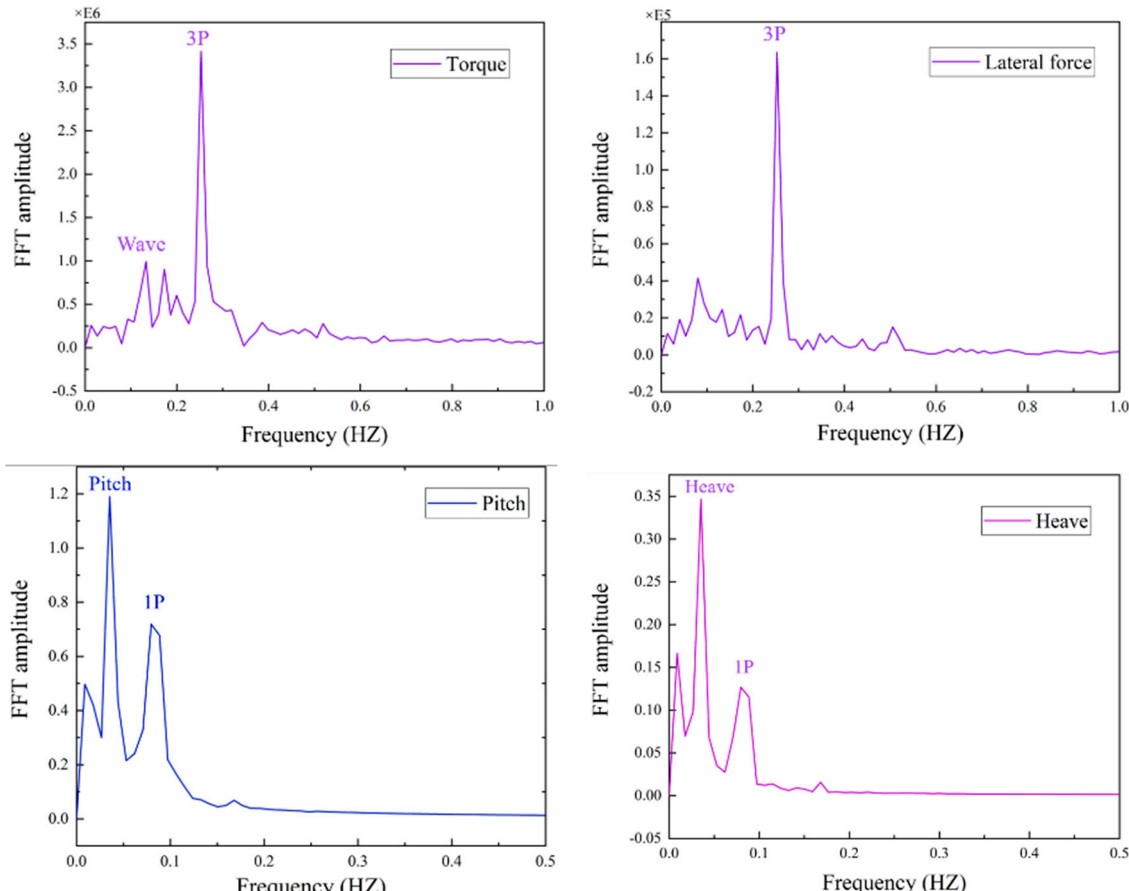

**Figure 12.** The frequency analysis results of aerodynamic loads and hydrodynamic motions.

The natural frequency of the floating system is 0.039 in pitch and 0.056 in heave motion. The frequency distribution of the pitch and heave motion in the frequency domain indicates that the maximum peak is approximately aligned with the corresponding natural frequency. The secondary peak frequency, approximately 0.08, is akin to the rotation frequency of the rotor. This finding suggests that the motion response of the floating VAWT system is primarily influenced by wind load and the natural period.

In summary, the aerodynamic load generated by the rotor is primarily influenced by wind, with a negligible contribution from waves. The motion responses of the entire system are mainly determined by its natural property, albeit wind load exerts a considerable impact on it as well.

Presented in Figure 13 are the stress and displacement distributions of the entire machine structure at a specific instant. The structural stress is primarily concentrated in the rotor-strut connecting zone, the root of the tower, the platform joint, and the lower floater of the platform. Concurrently, the displacement of the entire system is primarily concentrated in the rotor. The upper end of each blade and the top of the tower experience a larger displacement, indicating that the blades would undergo a vibration during rotation, and the tower exhibits lateral deflection. To gain a deeper understanding of the structural response characteristics of the complete system, it is crucial to analyze each component individually, including the blades, towers, and floating foundations. Consequently, time-domain and frequency-domain analyses are conducted for the stress of each gauge point and cross section, as shown in Figure 13.

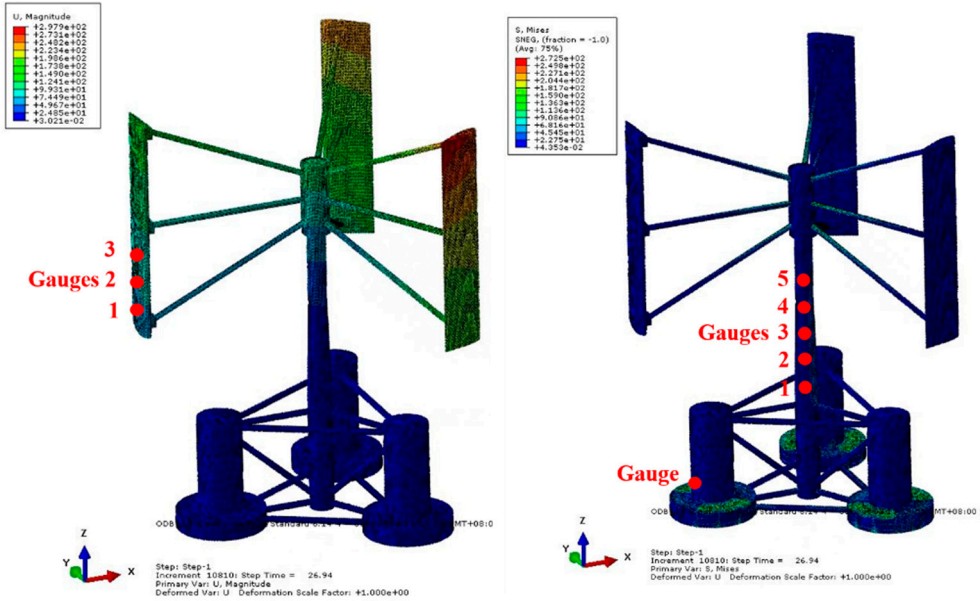

**Figure 13.** The stress and deformation distributions of the whole system in specific time T = 26.94 s.

The time-domain stress curve of a blade gauge, depicted in Figure 14a, exhibits a single peak per period. The frequency-domain analysis of three blade measuring points, as illustrated in Figure 14b, indicates a maximum peak corresponding frequency of approximately 0.0789, which is close to the rotation frequency. This finding suggests that the blade structural response is primarily associated with wind load. Additionally, the amplitude analysis reveals that the amplitude of measuring point 3 is the greatest, while that of measuring point 1 is the smallest. This suggests a higher stress concentration in the middle of the blade compared to the ends.

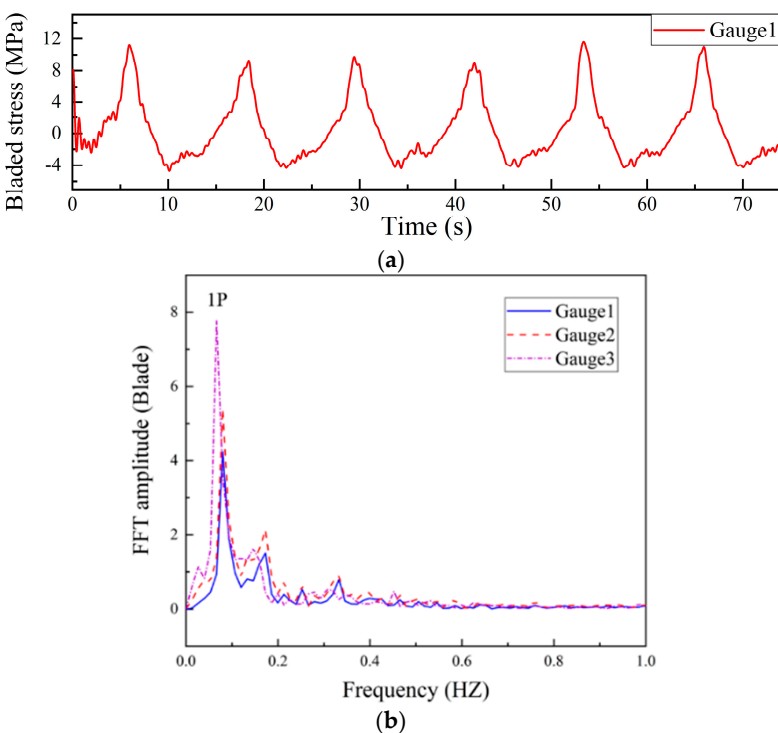

**Figure 14.** Structural response analysis of blade: (**a**) time-domain; (**b**) frequency-domain.

The stress levels of the five measuring points on the tower in both the time and frequency domains are depicted in Figure 15. It is evident that the stress amplitude stabilizes, but the absence of distinct periodicity is perplexing, necessitating a frequency domain analysis. Notably, the stress at the first measuring point exhibits four distinct peaks in the frequency domain, with the maximum peak corresponding to a frequency of approximately 0.3999, which is notably close to the first-order modal frequency, referring to tower 2-node vibration. The second largest peak corresponds to a frequency of approximately 0.252, which is closed to three times the rotation frequency (3P), suggesting that the tower periodically experiences the effects of three single blades. The third peak corresponds to a frequency of approximately 0.119, satisfying the wave frequency. This may be attributed to the relative movement between the upper structure and the lower platform, including rotation and translation, during the periodic wave action. Additionally, a slight peak appearing around 1.5 is closed to the third-order modal frequency, which relates to tower 3-node vibration. In summary, the tower is simultaneously influenced by wind and wave loads, which may lead to self-vibration of the structure, including 2-node and 3-node vibrations. Figure 15c depicts the maximum stress of the five evaluation points. It is notable that a significant reduction in stress is observed as the height increases. The maximum stress at the base of the tower approximates 35 MPa, whereas the maximum stress at the measuring point 5, which is situated closer to the upper section, reaches approximately 24 MPa.

The time-domain curve and frequency-domain distribution of the stress at the measuring point, which is situated on the upper buoy close to the column on the wave-facing side, are presented in Figure 16. The result indicates that the stress exhibits distinct periodicity and a stable amplitude, with the maximum stress level reaching approximately 55 MPa. The frequency-domain distribution reveals that the dominant frequency is approximately 0.121, which is related to the wave frequency. Additionally, the frequency of the second-largest peak is approximately 0.05, which is close to the natural frequency of the heave motion. This could be attributed to the periodic fluctuations in water pressure induced by periodic regular waves, coupled with the vertical motion, which also generates water pressure fluctuations. Consequently, the ultimate structural response of floating foundation is primarily contributed by these two sources.

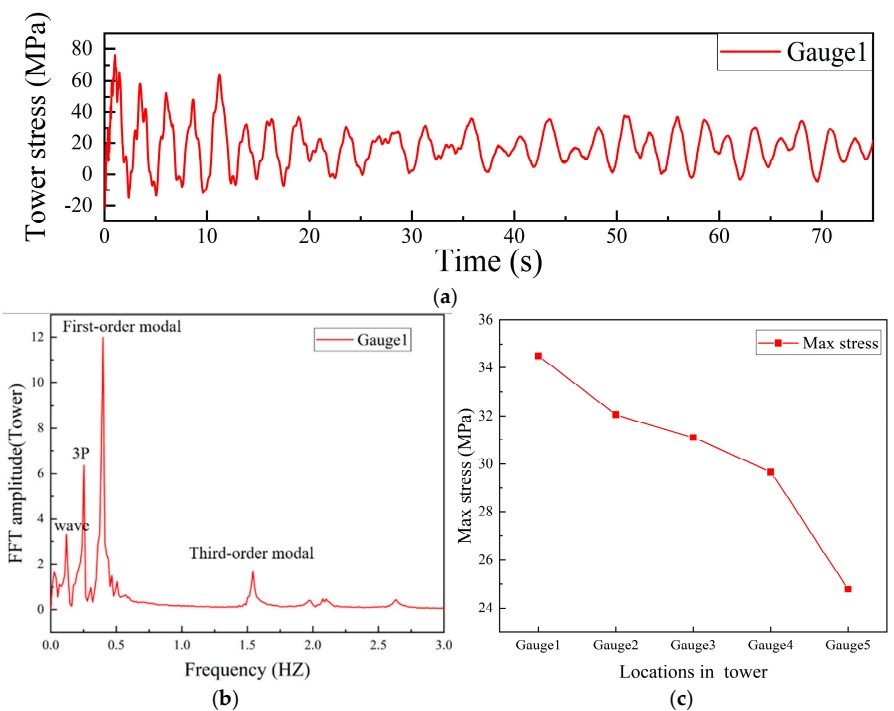

**Figure 15.** Structural response analysis of the tower: (**a**) time-domain; (**b**) frequency-domain; (**c**) statistics of maximum stress along the vertical direction.

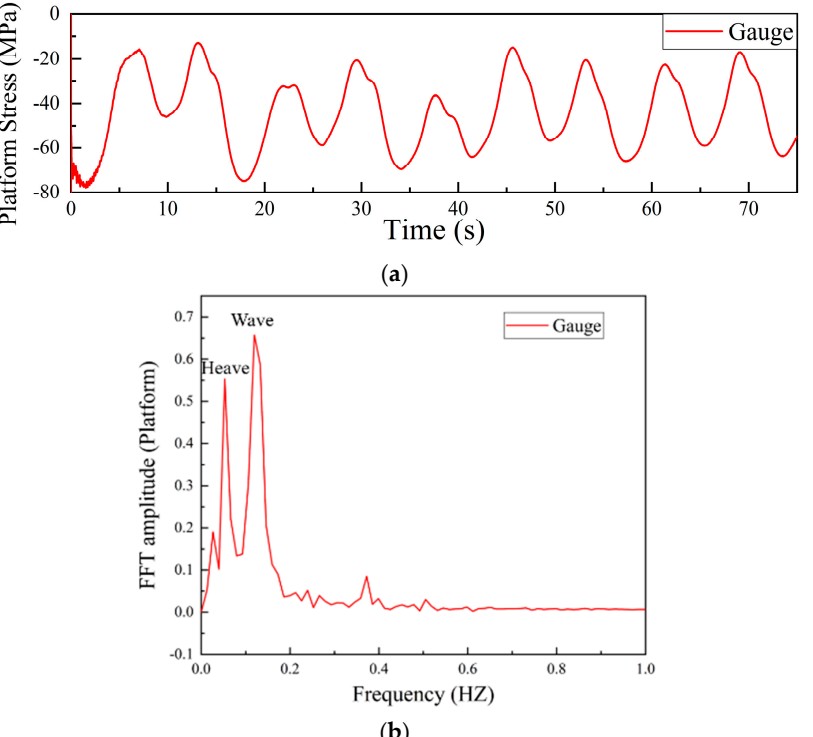

**Figure 16.** Structural response analysis of the floating foundation: (**a**) time-domain; (**b**) frequency-domain.

## 6. Discussion

The floating VAWT, situated in the deep sea, is frequently influenced by wind and waves. The wind speed, wave height, and misaligned angle are crucial parameters that contribute to the load, necessitating the examination of their effects on the floating VAWT.

Consequently, a statistical analysis focusing on the performance and structural safety of the entire machine is conducted.

### 6.1. Wind Speed

Figure 17a depicts the mean and standard deviation (STD) values of the total torque, axial thrust, and lateral thrust generated by the rotor at varying wind speeds. A significant increase in the mean value of axial thrust and STD is observed with the escalation of wind speed, with the mean value escalating nearly fivefold. The average lateral thrust remains relatively stable, exhibiting minimal variation, and is approximately zero; however, its STD experiences a significant increase. This suggests that although the average level of lateral thrust is consistently low, its fluctuation degree is even greater than that of axial thrust, which necessitates attention. A significant increase in the mean value and STD of the total torque is also observed with the increase in wind speed, with both increasing approximately sixfold.

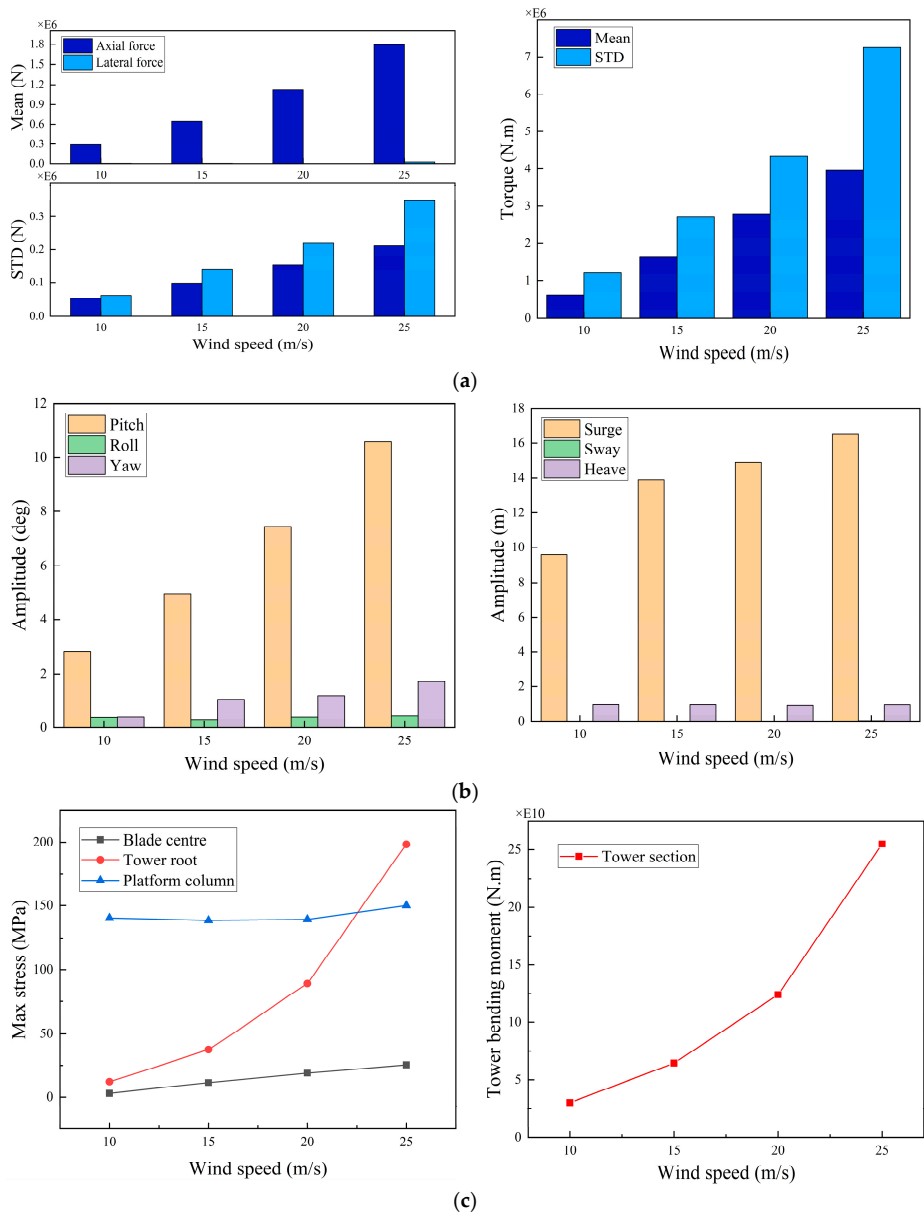

**Figure 17.** Aero-hydrodynamic performance in variable wind speed cases: (**a**) mean and STD value of thrust, lateral force, and torque; (**b**) amplitude of six-degree motion responses; (**c**) maximum structural responses of blade, tower, and platform.

Figure 17b demonstrates the maximum 6-DOF motion of the system under various wind speeds. As the wind speed elevates, the amplitudes of pitch, yaw, and surge escalate by 3.8 times, 4.2 times, and 1.7 times, respectively. However, the amplitudes of roll, sway, and heave demonstrate minimal changes. This could be attributed to the increase in axial force generated by the rotor with wind speed, as the rotor is situated in the top position. Consequently, rotation (pitch) and drift movements (surge) in longitudinal are induced. The alteration of the yaw angle of the entire system might be influenced by a violent disturbance of the lateral force.

The structural safety of each component of the entire machine is evaluated and compared, as illustrated in Figure 17c, which depicts the structural safety of each component, including the bending moment at the blade midpart, the tower root, and the platform column. It is evident that the maximum stress at the tower root demonstrates a nonlinear increase with increasing wind speed, becoming most prominent. At a wind speed of 25 m/s, the maximum stress approaches 200 MPa, and the bending moment at the tower root section exhibits a similar trend, almost five times. The maximum stress in the blade midpart also experiences a slight increase, from 5 MPa to approximately 25 MPa. The maximum stress at the platform column measuring point remains constant at around 145 MPa, with no apparent changes. Overall, the tower and blade structure exhibit greater sensitivity to changes in wind speed.

### 6.2. Wave Height

Figure 18a illustrates the influence of wave height on the floating VAWT. It is evident that the mean and STD values of the aerodynamic performance demonstrate minimal alterations with the increase in wave height. However, a slight increase in the STD of the total torque is observed. This might be attributed to the fact that as wave height elevates, the amplitude of motion in the entire system amplifies, thereby notably altering the wind energy utilization zone and the angle of incidence of the upper rotor. Consequently, the degree of disturbance in the total torque augments, but the overall mean level remains stable.

Figure 18b illustrates the maximum 6-DOF motion of the system under various wind speeds. With the increasing wave amplitude, the pitch, surge, and heave exhibit a significant enhancement, approximately multiplying by 1.5, 1.3, and 1.5 times, respectively. In contrast, the amplitudes of roll, yaw, and sway remain small and demonstrate no significant change. It is noticed that the variable wave cases belong to head wave condition, the floating foundation experiences rotation and drift in the wave propagation direction, and the platform undergoes periodic heave motion.

Figure 18c demonstrates the structural response variation of each component under different wave height. Different from the effect pattern induced by wind speed, the maximum stress on the blade and tower does not exhibit a significant change with wave height, remaining within a stable range. Nonetheless, as the wave height increases, the maximum stress of the platform column experiences a notable increase, from 50 MPa to 210 MPa. Despite the fact that the section bending moment at the tower root increases with wave height, reaching approximately 1.1 times the original value, the effect induced by wave height is significantly less than the impact of wind speed.

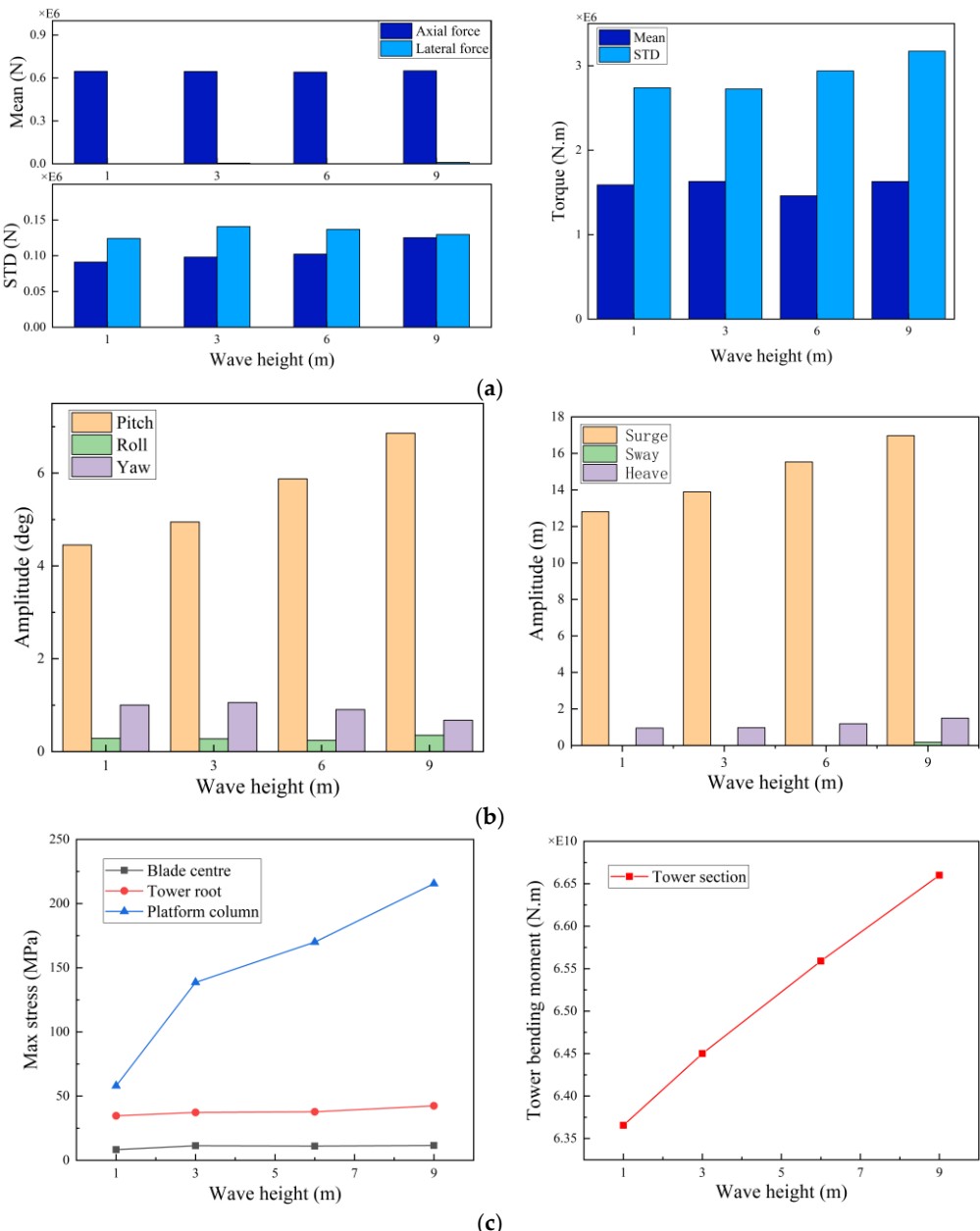

**Figure 18.** Aero-hydrodynamic performance in variable wave height cases: (**a**) mean and STD value of thrust, lateral force, and torque; (**b**) amplitude of six-degree motion responses; (**c**) maximum structural responses of blade, tower, and platform.

### 6.3. Misaligned Angle

The marine environment is inherently complex and dynamic, with wind and waves acting on the entire system from varying directions. Thus, it becomes imperative to examine the impact of wind-wave misaligned angles on the floating VAWT. To simulate various wind-wave misaligned angles, the wind inflow direction is kept constant, always facing the head-wave direction (positive *X*-axis), while the wave propagation direction gradually alters from the head-wave direction to the beam-wave direction (negative *Y*-axis) at an interval of 30 degrees. Consequently, four working conditions are established: 0°, 30°, 60°, and 90°.

Figure 19 shows the variation in aerodynamic and hydrodynamic performance of the floating VAWT. The fluctuations in the aerodynamic performance of the rotor generally diminish as the wind-wave misaligned angles increase, but this effect is less pronounced

compared to the influences of wave height and wind speed. Motion results reveal that the maximum motion of pitch and surge decreases with the increasing misaligned angle, whereas the maximum motion of roll and sway exhibits an opposite trend. Throughout the process of a wave transforming into a beam wave, the magnitude of the wave force impacting the floating platform remains virtually unaltered, yet the direction of its effect progressively shifts from longitudinal to horizontal. Consequently, the longitudinal movement decreases, while the lateral movement increases.

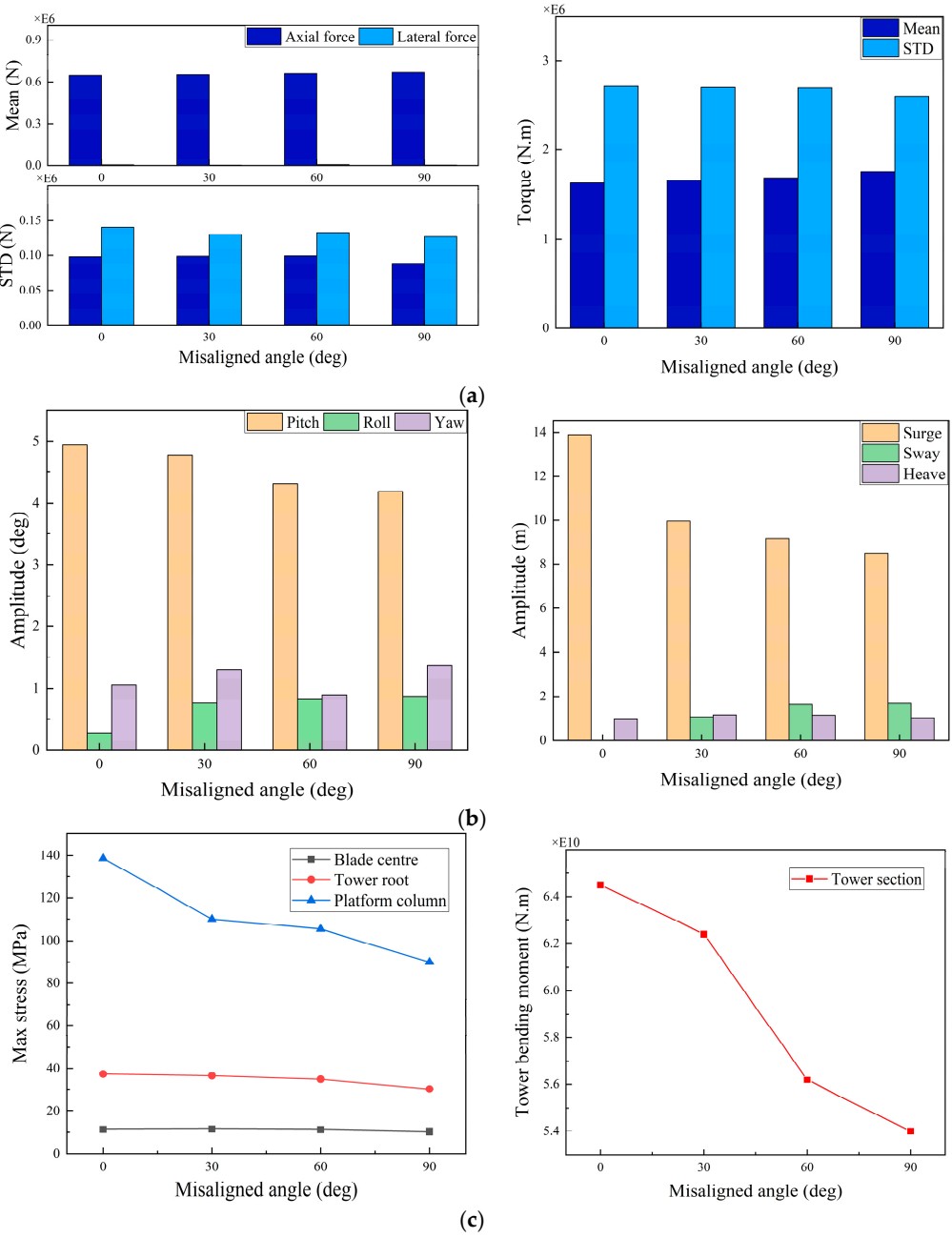

**Figure 19.** Aero-hydrodynamic performance in variable misaligned angle cases: (**a**) mean and STD value of thrust, lateral force, and torque; (**b**) amplitude of six-degree motion responses; (**c**) maximum structural responses of blade, tower, and platform.

The structural response changes are depicted in Figure 19c. It can be observed that the maximum stress in the platform column gradually decreases with the increasing misaligned angle, from 140 MPa to 90 MPa. The maximum stress at the tower root and the middle of the blade remains relatively stable, with the tower root at approximately 40 MPa and

the middle of the blade at approximately 10 MPa. The bending moment at the section near the tower root decreases by 15%. When the wave load shifts laterally, the column housing the gauge is no longer the first column in contact with the wave, resulting in a reduction in impact load. This might be the reason why platform stress is reduced. Additionally, due to the alteration in the direction of the wave load, the wind-induced torque and wave-induced torque superposition at the tower root is minimized, thereby decreasing the pressure exerted on the tower.

## 7. Conclusions

To obtain a more comprehensive and realistic load and structural response for floating VAWTs, this paper aims to extend CFD-FEA coupling technology into the floating VAWT area. An aero-hydro-moor-elastic fully coupled model is first developed. This model is validated by comparing results with several model experiments, including aerodynamics, hydrodynamics, and structure dynamics. Then, the geometrical and numerical models of a full-scale floating VAWT are introduced, and several variable wave height, wind speed, and misaligned angle cases are determined on the basis of ocean statistics. The load and structural response of a floating VAWT under typical operating conditions are investigated in both the time and frequency domains. The effects of wind speed, wave height, and misaligned angle on the floating VAWT are analyzed statistically. Key findings are presented as follows:

1.  CFD-FEA coupling technology is successfully extended to the field of floating VAWTs, and the fully coupled model developed is validated. Moreover, this method has the advantage of considering nonlinearities of aerodynamics and hydrodynamics, such as wake vortex and green water, etc., and the three-dimensional structures can be evaluated globally and locally. For instance, the findings of the flow distribution of the whole computational domain in Figure 11 and the stress and displacement distributions of the whole floating VAWT in Figure 13 are impossible to gain from the simplified fully coupled models.

2.  The analysis results of a typical operating condition reveal the primary contribution sources of the load and structural response for each part. The aerodynamic load is primarily influenced by the three-times rotation frequency (3P), while the system motion is associated with its natural frequency and wave frequency. The structural response of the blades is predominantly induced by the wind load, whereas the structural response of the tower is related to the 2-node mode, 3-node mode, wind load, and wave load. Lastly, the structural response of the platform is primarily associated with heave motion and wave frequency.

3.  The statistical analysis uncovers the susceptibility of various components within the floating VAWT to environmental parameters. Significant alterations in aerodynamic load levels are induced by wind speed, while changes in wave height and misaligned angle have a negligible impact on it. All three parameters result in considerable modifications to the hydrodynamic performance, particularly concerning pitch, roll, and surge. For the structural response, the tower structure is sensitive to wind speed and misaligned angle, blade structure is primarily dominated by wind speed, and the platform is more susceptible to wave height and misaligned angle.

In this paper, we conduct a fluid-structure interaction analysis of a straight-blade OF-VAWT. The investigation shows that the tower structural response is more complicated when it contains 2-node, 3-node, wind frequency, and wave frequency components. Further, statistical analysis identifies that the tower structure is sensitive to both wind speed and misaligned angle. This study provides a solution with higher precision for predicting structural safety in the detailed design stage, and the findings have implications for engineering design. However, the present fluid-structure interaction work concentrates on elastic deformation, and the fatigue as well as the plasticity are not yet considered; thus, related damage research will be conducted in the future. Meanwhile, it should be noted that the present CFD-FEA is time-consuming despite being precise in predicting detailed

structural problems. Hence, a simplified model with satisfying accuracy is also essential for initial designs.

**Author Contributions:** W.L. (Wenping Luo): simulation, validation, writing—original draft, visualization; W.L. (Weiqin Liu): funding acquisition, supervision, methodology, writing—review and editing; S.C.: software, methodology; Q.Z.: validation, methodology; X.S.: supervision, methodology. All authors have read and agreed to the published version of the manuscript.

**Funding:** This research was funded by the National Natural Science Foundation of China (No.: 52071243, 52101371) and the National Defense Basic Research Program of China (No.: JCKY2020206B037).

**Institutional Review Board Statement:** Not applicable.

**Informed Consent Statement:** Not applicable.

**Data Availability Statement:** Data are contained within the article.

**Conflicts of Interest:** The authors declare no conflicts of interest.

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
