# Peer review of "Development and Application of an FSI Model for Floating VAWT by Coupling CFD and FEA"

_jmse, doi:10.3390/jmse12040683_

Round 1
Reviewer 1 Report
Comments and Suggestions for Authors
In this article, the authors have developed an aero-hydro-moor-elastic fully-coupled model for a floating vertical axis wind turbine (VAWT) to predict the load and structural responses.
The paper addresses a very interesting topic, especially in the fact that it meets the increasing demand for research towards floating wind energy.
I suggest the authors discuss the effect of lateral deflection of the blade (if any) due to the wind loads in the dynamics.
I suggest the authors discuss the software used for the simulations.
I would recommend the authors describe the limitations of the study.
Author Response
Q1:I suggest the authors discuss the effect of lateral deflection of the blade (if any) due to the wind loads in the dynamics.
A1: Thanks for your suggestions. The main object of this paper is to verify the feasibility of the newly developed model, and briefly analyze the dominant structural response characteristics. Your idea is impressive and of great interest to investigate in future such as detailed rotor research.
Q2:I suggest the authors discuss the software used for the simulations.
A2:Thanks for your advice. The STARCCM and ABAQUS are integrated in this paper, and some descriptions are added in section2.1.
Q3:I would recommend the authors describe the limitations of the study.
A3: Thanks for your recommendation. The significant limitation of this CFD-FEA method is described at the end of paper. ‘Meanwhile, it should be noted that present CFD-FEA is time-consuming despite it being precise in predicting detailed structural problems. Hence, a simplified model with satisfying accuracy is also essential for initial designs’
Reviewer 2 Report
Comments and Suggestions for Authors
The paper presents a direct CFD-FEA coupling method to predict the hydro-elastic performance of floating Vertical axis wind turbines. The analysis uses URANS technique or solving Navier-Stokes equations. The FE analysis is restricted to the elastic region. Waves are included in the analysis.
It seems that commercial packages have been used in solving the hydro- & aerodynamic problem as well the structural one. Some details about the solvers used their basic parameters and their co-operation should be provided. How do they cope with the effect of irregular waves?
Following the remedy of these shortcomings, I recommend this paper to be published.
Comments on the Quality of English LanguageMinor editing corrections.
Author Response
Q1:It seems that commercial packages have been used in solving the hydro- & aerodynamic problem as well the structural one. Some details about the solvers used their basic parameters and their co-operation should be provided.
A1:Thanks for your advice. The STARCCM and ABAQUS are integrated in this paper, and some descriptions are added in section2.1.
Q2:How do they cope with the effect of irregular waves?
A2: Thanks for your question. It is acknowledged that CFD simulation is time-consuming, despite the irregular waves could be generated theoretically. Thus, this CFD-FEA method is mainly used for detailed design such as extreme conditions which are equal to the worst from irregular wave statistics.
Reviewer 3 Report
Comments and Suggestions for Authors
45-48: Why would VAWT have significantly different dynamic behaviour and what is the significance of this?
83: Define soft blades.
143, 207: The authors reference previous study [18]. It would be good to expain in all sections the novelty of the current study.
Figure 4: Can the authors explain differences in published results and present study at TSR = 2.4 and azimuth angles of 0, 30 and 90 deg?
Author Response
Q1:45-48: Why would VAWT have significantly different dynamic behaviour and what is the significance of this?
A1: Thanks for your questions. The VAWT has two zones named up-wind and down-wind, which means two interactions are produced between inflow and blade, and the attack angle varies all the time during rotation. However, present HAWT type only contains up-wind, and the attack angle is constant unless controller works. That’s why the aerodynamic performance is obviously different. Meanwhile, the structural arrangement connecting blade and main tower also differ, cantilever beam for HAWT but simply supported beam for most VAWT. A simplified description is added in article.
Q2:83: Define soft blades.
A2:Thanks for your suggestion. The description ‘low stiffness’ is added in article.
Q3:143, 207: The authors reference previous study [18]. It would be good to explain in all sections the novelty of the current study.
A3: Thanks for your advice. The additional supply sentence is added in corresponding position.
Q4:Figure 4: Can the authors explain differences in published results and present study at TSR = 2.4 and azimuth angles of 0, 30 and 90 deg?
A4: Thanks for your revision. First, the published experiment data is smaller than both present and published IDDES results, this might be induced by additional human or mechanical error as simulations are completed in ideal environment. Second, these two simulations have satisfied coincidence in both amplitude and phase. Third, the differences you mentioned in partial conditions, such as TSR=2.4, the azimuth angle of 0, 30, 90, might be caused by the simulation model, while IDDES for published one and URANS for present. Generally, the IDDES model is more capable of capturing eddy while URANS focus on mean variation. In a word, these tiny differences are acceptable.
Reviewer 4 Report
Comments and Suggestions for Authors
The proposed paper “Development and application of an FSI model for Floating VAWT by coupling CFD and FEA” reports the application of a multi-model system for the analysis of a vertical axis wind turbine installed offshore.
The manuscript is very interesting, reporting a set of new results regarding this kind of turbine. Minor correction are necessary for its publications:
1. Avoid acronyms in the title and abstract
2. At the beginning section 2 the structure of the domain should be introduced, for a proper description (and ease understanding) of the system coupling: e.g. the rotation domain is not defined and the overset method is not accurately described
3. The novelty of the method with respect to reference [31] should be clearly declared. The methods appears to be the same. Also for this reason, the methodological section should be summarized, removing all the text and figures which overlaps with references [18] and [31].
4. Lines152-154: not clear sentence. What is the “press of time”? Is it necessary to write that the Reynolds stress tensor is a second order tensor?
5. There are symbols used more than once, with different meanin (e.g. alpha and beta), and some simbols are not defined (e.gg. c in equations 12-17)
6. Lines269-272: not clear, please explain better. What is the experimental reference of this sentence?
7. In the conclusion, emphasize the advantage of your results with respect to the results that can be obtained with less time-consuming models
8. Since you are using strongly nonlinear models (FEM and CFD), why don’t you discuss highlight nonlinear processes, which could be the source of the obtained results?
Comments on the Quality of English LanguageLines152-154: not clear sentence. What is the “press of time”? Is it necessary to write that the Reynolds stress tensor is a second order tensor?
Author Response
Q1. Avoid acronyms in the title and abstract
A1:Thanks for your suggestion. For the title, FSI, VAWT, CFD and FEA are used. We are willing to employ the full name. However, the title would be “Development and application of a Fluid-Structure Interaction model for Floating Vertical Axis Wind Turbines by coupling Computational Fluid Dynamics and Finite Element Analysis”. To avoid the redundancy, some acronyms are used.
Q2. At the beginning section 2 the structure of the domain should be introduced, for a proper description (and ease understanding) of the system coupling: e.g. the rotation domain is not defined and the overset method is not accurately described
A2: Thanks for your advice. The descriptions are supplied and added at the beginning section 2.
Q3. The novelty of the method with respect to reference [31] should be clearly declared. The methods appears to be the same. Also for this reason, the methodological section should be summarized, removing all the text and figures which overlaps with references [18] and [31].
A3: Thanks for your suggestion. It should be noted that the difference between these three studies([18],[31] and present). [18] is the author’s first attempt to couple CFD and FEA to solve plastic problems(ship as example), [31] is the initial research for floating VAWT which only considered the aero-hydro-mooring loads and the structure is not included due to difficulties in coupling technology. The present study breaks through the CFD-FEA coupling technology in floating VAWT application. Therefore, the novelty is remarkable despite some fundamental theory being similar, and some description about these is added at beginning of section 2.
Q4. Lines152-154: not clear sentence. What is the “press of time”? Is it necessary to write that the Reynolds stress tensor is a second order tensor?
A4: Thanks for your correction. The press of time is a mistake which should be ‘pressure’. And the description ‘Reynolds stress tensor is a second order tensor’ is needless.
Q5. There are symbols used more than once, with different meaning (e.g. alpha and beta), and some symbols are not defined (e.gg. c in equations 12-17)
A5:Thanks for your check. The same symbol used multiple times is corrected, and the c in equations 12-17 is defined by Eq. 14.
Q6. Lines269-272: not clear, please explain better. What is the experimental reference of this sentence?
A6: Thanks for your check. The sentence is corrected and some needless word is deleted.
Q7. In the conclusion, emphasize the advantage of your results with respect to the results that can be obtained with less time-consuming models
A7: Thanks for your advice. The advantages in capturing nonlinearity of present model are added in the first conclusions.(eg. For instance, the flow distribution of whole computational domain in Fig. 11, and the stress as well as displacement distributions of whole floating VAWT in Fig. 13, these finds are impossible from those simplified fully coupled models.)
Q8. Since you are using strongly nonlinear models (FEM and CFD), why don’t you discuss highlight nonlinear processes, which could be the source of the obtained results?
A8: Your comments are pretty impressive. The present coupled method can capture nonlinearity. Whereas, this paper mainly focuses on the development and conducts a simple case study for application. In fact, a related study, standing on the nonlinearity effects, started after developing this method. We believe this research would be interesting and published future.